# The p53 protein is a suppressor of Atox1 copper chaperon in tumor cells under genotoxic effects

**Sergey Tsymbal**[1‡]*, **Aleksandr Refeld**[1], **Viktor Zatsepin**[2], **Oleg Kuchur**[1‡]

**1** International Institute 'Solution Chemistry of Advanced Materials and Technologies', ITMO University, St. Petersburg, Russia, **2** Kirov Military Medical Academy, St. Petersburg, Russia

‡ ST and OK are joint senior authors on this work.
* zimbal@scamt-itmo.ru

**Data Availability Statement:** All relevant data are within the paper and its Supporting Information files.

**Funding:** This research was funded by Russian Science Foundation, grant number 22-24-0058.

## Abstract

The p53 protein is crucial for regulating cell survival and apoptosis in response to DNA damage. However, its influence on therapy effectiveness is controversial: when DNA damage is high p53 directs cells toward apoptosis, while under moderate genotoxic stress it saves the cells from death and promote DNA repair. Furthermore, these processes are influenced by the metabolism of transition metals, particularly copper since they serve as cofactors for critical enzymes. The metallochaperone Atox1 is under intensive study in this context because it serves as transcription factor allegedly mediating described effects of copper. Investigating the interaction between p53 and Atox1 could provide insights into tumor cell survival and potential therapeutic applications in oncology. This study explores the relationship between p53 and Atox1 in HCT116 and A549 cell lines with wild type and knockout *TP53*. The study found an inverse correlation between Atox1 and p53 at the transcriptional and translational levels in response to genotoxic stress. Atox1 expression decreased with increased p53 activity, while cells with inactive p53 had significantly higher levels of Atox1. Suppression of both genes increased apoptosis, while suppression of the *ATOX1* gene prevented apoptosis even under the treatment with chemotherapeutic drugs. The findings suggest that Atox1 may act as one of key elements in promotion of cell cycle under DNA-damaging conditions, while p53 works as an antagonist by inhibiting Atox1. Understanding of this relationship could help identify potential targets in cell signaling pathways to enhance the effectiveness of combined antitumor therapy, especially in tumors with mutant or inactive p53.

## Introduction

With the accumulation of data on the antitumor effects of radio- and chemotherapy, numerous attempts have been made to identify the molecular mechanisms of survival and death of malignant cells. One of the most obvious markers, whose history began more than 40 years ago, is the oncosuppressor p53. This protein is a crucial regulator of tumor survival and death, an inducer of apoptosis, reparative processes, and also plays an important role in cell response to ROS damage [1–4]. Furthermore, the balance of redox reactions in the cell is closely linked to the regulation of intracellular homeostasis of transition metals such as zinc (Zn), iron (Fe),

The funders had no role in study design, data collection and analysis, decision to publish, or preparation of the manuscript.

**Competing interests:** The authors have declared that no competing interests exist.

**Abbreviations:** ROS, Reactive oxygen species; CDKN1A, cyclin-dependent kinase 1A inhibitor (TP21); CCND1, Cyclin D1 gene; WT, wild type cells; KO, cells with TP53 gene knockout; PMA, phorbol-12-myristate-13-acetate.

and copper (Cu) [5–10]. However, there is limited information available regarding the correlation or codependence between the expression levels of p53 and proteins involved in metal metabolism in tumors [11–14]. Given the unique properties of copper and copper-binding proteins, investigating the metabolism of this metal becomes particularly attractive for developing approaches to combined tumor therapy [15, 16]. Copper plays a crucial role in redox reactions and the elimination of ROS, as it is an integral part of the superoxide dismutase enzyme [17]. Additionally, copper can influence the level of intracellular glutathione, a major antioxidant molecule in cells [18]. Despite these important functions, the understanding of copper's involvement in the occurrence and progression of tumor diseases is still in its early stages. Recent studies have focused on the dysregulation of copper-associated metallochaperones and enzymes during oncogenesis, as well as their potential therapeutic applications [19, 20]. Notably, enhancing the antitumor activity of disulfiram through the addition of copper ions has shown promising effects [21, 22]. Moreover, research conducted at the Laboratory of Diagnostics and Targeted Radiopharmaceutical Therapy of the University of Wisconsin has demonstrated a decrease in copper transport into the nucleus when p53 expression is inhibited or absent [23]. Further investigations have revealed a correlation between copper ion concentration and the activity of signaling cascades associated with malignancy, such as B-Raf, Akt, and HIF1 [24]. Inhibition of various copper carriers or chelation of copper ions also affects corresponding pro-oncogenic signaling pathways, including ERK, MAPK, NF-kB, and EGFR/Src/VEGF, which are involved in angiogenesis [25–28]. These findings suggest an association between p53 and copper-dependent proteins in tumor progression, highlighting the involvement of this tumor suppressor in the regulation of copper metabolism.

Considering our knowledge of the importance for oncotherapy of such copper-associated proteins as SOD1, CTR1, and angiogenin [5, 6, 29–31], an equally important player in copper metabolism, the Atox1 chaperone, which is an antioxidant and a transcription factor, remains aside. The role of this protein in tumor responses to genotoxic effects was unclear until recently. Only in 2015, a group of scientists from the Department of Hematology and Medical Oncology at Emory University showed that inhibition of Atox1 directly reduces the proliferation of tumor cells [32], and the binding of Atox1 to the cis-element of Cyclin D1 stimulates the growth and proliferation of mouse embryonic fibroblasts, as well as SW480 and SW620 colorectal cancer cells [33, 34]. Atox1 is also shown to influence DNA repair by transcriptionally activating the MDC1 protein [35]. Knockdown of Atox1 in non-small cell lung cancer cells reduces proliferative and growth processes [36]. Apparently, p53 activation, depending on the cell line and type of exposure, can differently affect the expression of Atox1, the induction of which protects the cell from death under ionizing radiation and cytotoxic drugs by eliminating ROS [37].

As a result, the data available in the literature on this topic are limited and rather contradictory. However, the general trend towards the study of copper metabolism and its relationship with typical cancer markers is very clear. We continue this trend, but our goal is to elucidate the role of the p53 tumor suppressor in the regulation of one particular participant in the copper metabolism pathways, Atox1, by paying attention to the responses of this protein to typical tumor therapy stimuli, such as cytotoxic drugs and ionizing radiation. The data will lay the foundation for further research on this topic and the possible implementation of the development of new anticancer drugs.

## Results

### Atox1 activity is increased in cells with the *TP53*$^{-/-}$

At the first stage, we assessed the basic level of gene expression and induction of the Atox1 protein in HCT116 colorectal cancer and A549 lung carcinoma cell lines with the wild type (WT)

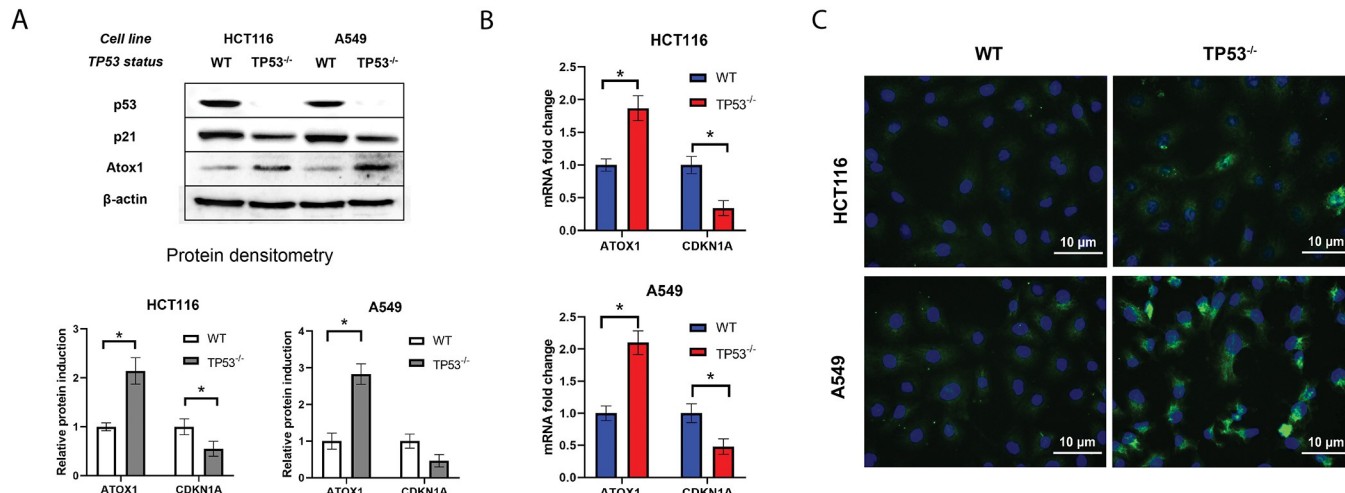

**Fig 1. Dependence of Atox1 and p53 levels in HCT116 and A549 cell lines with different TP53 status.** A—immunoblotting with antibodies to p53, p21, and Atox1; β-actin was used as a normalization. A densitometric analysis of the obtained data is shown below. B–RT-qPCR analysis with primers for *TP53*, *CDKN1A*, and *ATOX1* genes; *GAPDH* gene was used as a reference. C—immunofluorescence staining with primary antibodies to Atox1 and secondary antibodies with AlexaFluor488. DAPI was used for nuclei staining. WT–wild type cells, TP53$^{-/-}$–cells without TP53. For all experiments: n = 3, mean +/− SEM, paired Student t-test, p < 0.05.

or inactivated by the CRISPR-Cas9 tumor suppressor gene TP53 (*TP53$^{-/-}$*). Immunoblotting analysis revealed that cells with functional p53 exhibited reduced Atox1 activity, whereas p53 knockout cells showed a significant increase in Atox1 protein content by approximately 2.2–2.8 times under normal conditions. This trend was observed in both cell lines, with no statistically significant difference in the baseline levels of Atox1 protein between the two lines (Fig 1A). To further validate our findings, we used p21, a p53-dependent inhibitor of cyclin-dependent kinase 1, as an additional control. Accumulated p53 activates the *CDKN1A* gene, leading to cell cycle arrest at the G$_2$/M phase and inhibition of Cdc25, thereby facilitating DNA repair processes [38]. Consequently, the level of p21 decreases when p53 is suppressed. In p53 knockout cells, the amount of p21 protein was found to be reduced by approximately 2-fold compared to the control (Fig 1).

We also examined the transcriptional regulation of the Atox1 gene in relation to *TP53* status. Real-time PCR analysis revealed that the relative expression of *ATOX1* mRNA was approximately 2–2.5 times higher in p53 knockout cells compared to wild-type cells (normalized to 1.0), while the expression of *CDKN1A*, which encodes p21, decreased by approximately 2-fold (Fig 1B).

Furthermore, immunofluorescence microscopy allowed us to visualize the observed pattern of increased Atox1 activity in cells with *TP53* inactivation. The A549 cell line exhibited approximately 2.5 times higher levels of the metallochaperone compared to the HCT116 cell line (Fig 1C). However, this method did not enable the detection of Atox1 translocation into the nucleus as a transcription factor [37], in connection with this, a method of subcellular fractionation with analysis of Atox1 distribution by immunoblotting was subsequently proposed.

These findings raise important questions regarding the role of Atox1 as a p53-dependent factor, which is known to be a crucial sensor for responses to DNA damage, cell cycle regulation, and repair processes. Specifically, it prompts us to investigate whether Atox1 activity is altered in response to cytostatic and cytotoxic effects and whether it contributes to the regulation of the survival-death balance in tumor cells, particularly those harboring p53-null mutations.

## Atox1 is induced in a p53-dependent manner during genotoxic stress

According to Beaino W. et al., the Atox1 protein is induced in a p53-dependent manner in response to the cytotoxic drug cisplatin [37]. According to the authors, this can be explained by the ability of cisplatin to a certain extent to replace copper ions and bind to Atox1, acting as a cofactor, which induces this protein. However, given the ability of Atox1 to act as a transcription factor and play a role in the processes of response to external stimuli [33], we formulated two hypotheses: 1) expression of Atox1 increases in response to various genotoxic signals (cytotoxic drugs, ROS inducers, ionizing radiation) at the level of transcription and translation; 2) this induction is p53-dependent.

Indeed, as demonstrated above and will be further discussed, Atox1 expression is elevated in TP53 knockout sublines (HCT116TP53$^{-/-}$, A549TP53$^{-/-}$), which contradicts previous findings by Beaino et al., who only observed Atox1 induction in HCT116 WT, while in cells with TP53$^{-/-}$ the level of Atox1 decreased by nearly 2-fold [37]. This study did not replicate the effect in the MEF mouse fibroblast line. In contrast, we present a sequential pattern of Atox1 activation in HCT116 and A549 cell lines with inactivated TP53 both at the transcriptional and translational levels.

Returning to the first hypothesis, *ATOX1* expression is clearly upregulated in response to multiple genotoxic stimuli. Thus, p53 reacts similarly to all genotoxic agents, except for hydrogen peroxide, both at the mRNA and protein levels, which is quite expected given its role in the response to DNA damage [39], which correlated with the results of our experiments (Fig 2). At the same time, in the HCT116TP53$^{-/-}$ and A549TP53$^{-/-}$ cell lines, Atox1 is activated upon the addition of 0.1μM doxorubicin, 80nM PMA, and 10μM bleomycin (and, to a certain extent, 35μM cisplatin in wild-type cells) and is weakly activated when p53 is normally functioning. Note that PMA is not a cytotoxic drug, but an activator of protein kinase C (PKC) [40]. PKC is found to activate many signaling pathways, including NF-kB [41, 42] and MAPK [43]. Both pathways were shown to interact with Atox1 [44, 45], furthermore, PKC could directly phosphorylate Atox1 [46]. Therefore, of particular interest was the analysis of the effect of PMA on Atox1 activity depending on p53 status.

Thus, in HCT116 cells with suppressed p53, Atox1 induction in response to cisplatin, doxorubicin, PMA, and bleomycin was 2.1, 2.5, 3.0, and 2.8 times higher compared to the control, respectively. For the A549TP53$^{-/-}$ line, these values were 2.0, 2.8, 2.4, and 2.2 times, respectively (Fig 2A). The weak response to H$_2$O$_2$ is apparently associated with its short half-life/rapid decay, and rapid cellular responses that regulate changes in the cell in response to oxidative stress, which do not lead to large-scale translational responses. It is worth noting that Atox1 inducibility is on average similar for both cell lines to the respective drugs, which was not observed in previous studies. Doxorubicin, PMA, and bleomycin caused the strongest differences in Atox1 expression in both lines and were therefore selected for PCR analysis.

Expression analysis of the *TP53*, *CDKN1A*, and *ATOX1* genes upon exposure to previously selected drugs confirmed the data obtained by immunoblotting (Fig 2B). The control (intact cells, no effects) was taken as 1.0. The addition of doxorubicin to HCT116 and A549 cells led to an increase in the expression of *TP53* and *CDKN1A* by 3–4 and 4–5 times, respectively, relative to the control, for PMA these values were equal to 3–4 for both genes relative to the control, respectively, for bleomycin—5–6 in both cases. At the same time, Atox1 expression for all compounds did not exceed a 2-fold change in the studied cell lines with wild type p53. In the case of *TP53* knockouts, the *CDKN1A* gene was practically not expressed, and Atox1 activity increased to ~3.5 change fold when exposed to doxorubicin and PMA (both cell lines) and up to 4.5-fold when bleomycin was added (Fig 2B). In general, transcriptional and translational response data for chemotherapeutic agents were comparable.

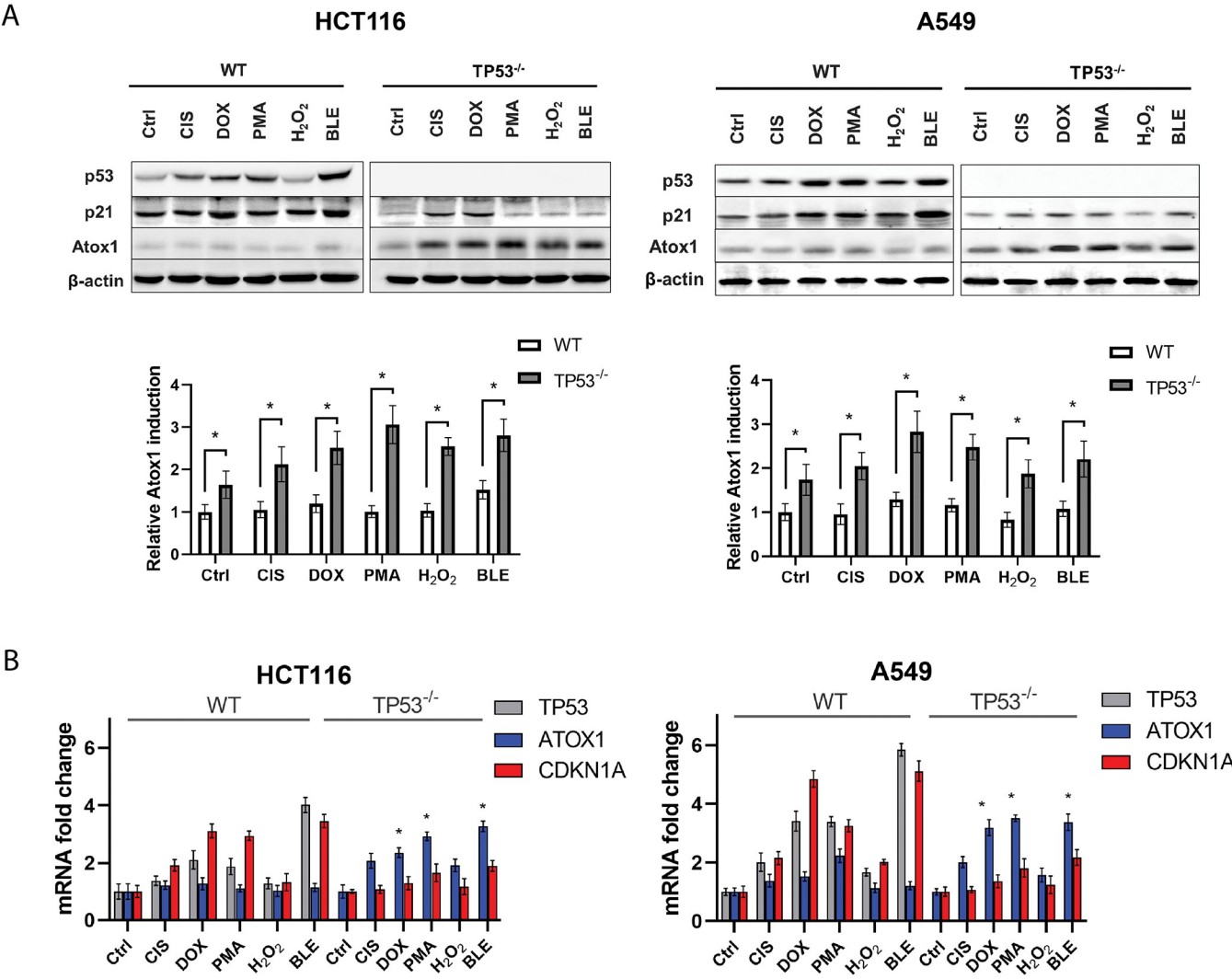

**Fig 2. Influence of cytotoxic agents on the activity of Atox1 at different status (WT and KO) of the *TP53* gene in A549 and HCT116 cell lines, 24h after drugs exposure.** A—immunoblotting with antibodies to p53, p21, and Atox1; β-actin was used as a normalization. A densitometric analysis of the obtained data is shown below. B–RT-qPCR analysis with primers for *TP53*, *CDKN1A*, and *ATOX1* genes; *GAPDH* gene was used as a reference. DOX–doxorubicin (0,1μM), CIS–cisplatin (35μM), PMA–phorbol-12-myristate-13-acetate (80nM), $H_2O_2$ –hydrogen peroxide (450μM), BLE–bleomycin (10μM). WT–wild type cells, TP53$^{-/-}$ –cells without *TP53*. For all experiments: n = 3, mean +/− SEM, two-way ANOVA, $p < 0.05$.

It is generally accepted that the main role of Atox1 lies in its functions as a transcription factor under stressful conditions [47]. Atox1 has been shown to has a nuclear localization sequence and migrate into the nucleus under the action of cisplatin in a p53-dependent manner [37]. To assess the nuclear translocation of Atox1, immunocytochemical staining of the Atox1 protein was performed under the influence of doxorubicin (0.1μM), PMA (80nM), and bleomycin (10μM), which showed good results in transcriptional and translational activation. Fluorescence microscopy showed no discernible nuclear translocation of Atox1 upon genotoxic exposure, with the protein increasing markedly, especially with doxorubicin and bleomycin (Fig 3A). It is likely that the drugs used in this experiment, unlike cisplatin, have a weak ability to bind Atox1 and induce its migration into the nucleus, since they do not interact with copper metabolism proteins [48]. Additionally, we proved that expression of Atox1 could be induced by PMA addition, because this protein is bound to the pathways that are induced by PKC.

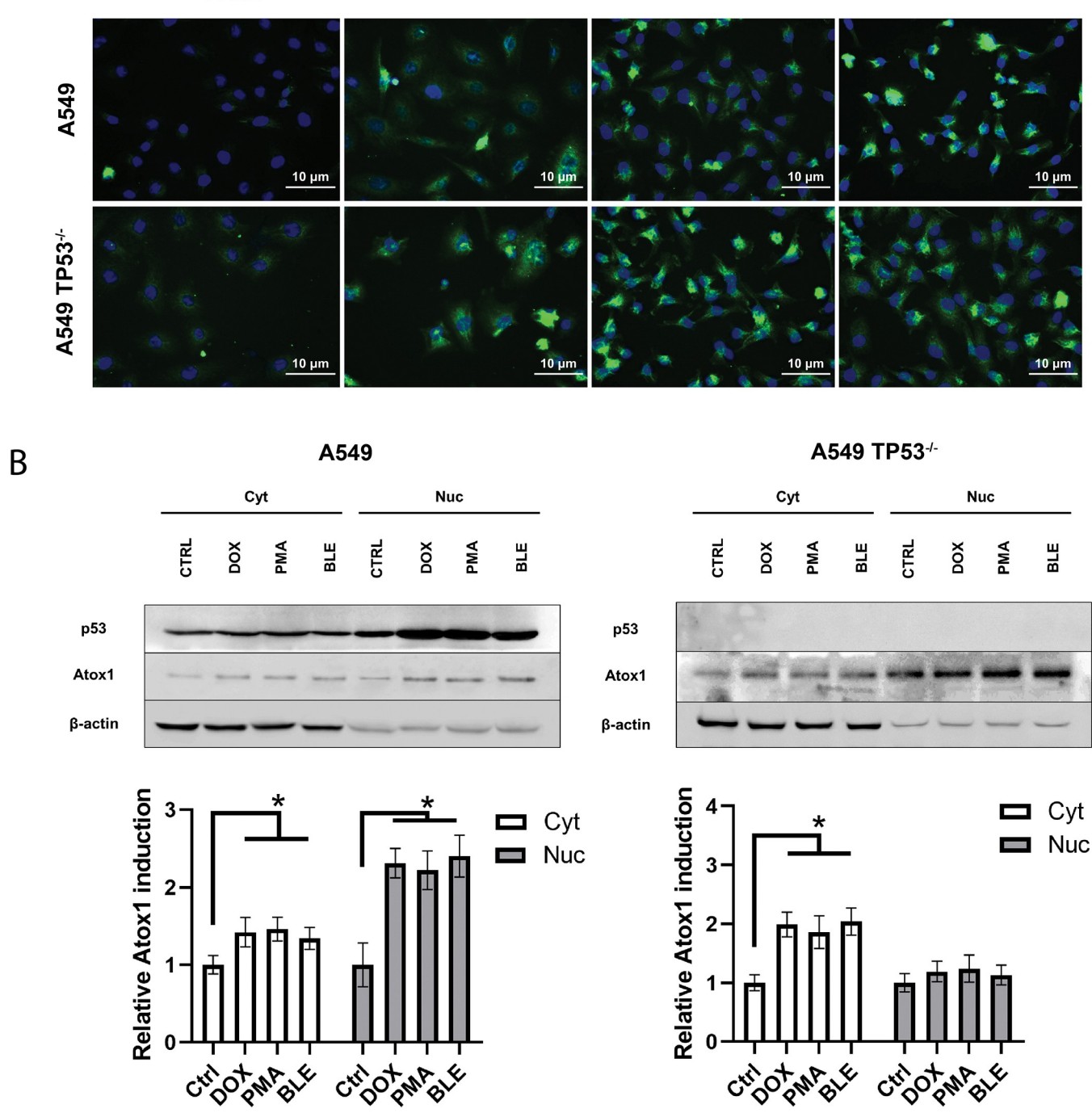

**Fig 3. The effect of cytostatic agents on the intracellular localization of the Atox1 protein at different statuses (WT and KO) of the TP53 gene in the A549, 24 hours after drugs exposure.** A—immunofluorescence staining with primary antibodies to Atox1 and secondary antibodies with AlexaFluor488. DAPI was used for nuclei staining. B–immunoblotting with antibodies to p53 and Atox1 after subcellular fractionation; β-actin was used as a normalization (Cyt—cytoplasmic and Nuc—nuclear fractions). A densitometric analysis of the obtained data is shown below. DOX–doxorubicin (0,1µM), PMA–phorbol-12-myristate-13-acetate (80nM), BLE–bleomycin (10µM). WT–wild type cells, TP53$^{-/-}$–cells without *TP53*. For all experiments: n = 3, mean +/− SEM, two-way ANOVA, p < 0.05.

To further confirm the redistribution of metallochaperone into the nucleus, the subcellular fractionation method was used, which made it possible to divide the samples into two fractions—nuclear and cytoplasmic, and to more strictly assess the distribution of the protein in the cell (Fig 3B). While for p53 there was a significant increase in the protein level by 3–4 times in the nucleus upon its genotoxic activation, the translocation of Atox1 from the cytoplasm into the nucleus in A549 WT cells was less pronounced—approximately 2 times when exposed to doxorubicin, PMA and bleomycin. In the case of cells with inactivated p53, the migration of Atox1 from the cytoplasm increases 3–3.5 times after treatment with doxorubicin, PMA and bleomycin, but only relative to the wild-type control. At the same time, the amount of protein in untreated A549p53KO cell line and when exposed to drugs is practically no different. These data indirectly indicate the role of Atox1 as the transactivator in the absence of the normal TP53 gene.

In addition to cytotoxic drugs exposure, we investigated the impact of ionizing radiation on the transcriptional and translational responses of Atox1. Ionizing radiation is known to cause single- and double-strand DNA breaks and generate reactive oxygen species through water radiolysis [49, 50]. To generate gamma radiation, we utilized the RUM-17 radiotherapy unit with an effective therapeutic dose of 10 Gray (Gy).

Our findings corroborate previous observations on the response of p53 to radiation. Specifically, in HCT116 cells, p53 activity increased by three-fold compared to the non-irradiated control, and in A549 cells, it increased by 3.6-fold. Similarly, the induction of the p21 protein followed a similar pattern, albeit with lower activity levels. The expression of p21 was ~2 times higher than the control values, and its induction was reduced in cell lines with inactive TP53 but increased upon irradiation.

In contrast, the metal chaperone Atox1 exhibited minimal response to gamma radiation, irrespective of the p53 status. However, in irradiated A549TP53$^{-/-}$ cells, there was a slight suppression of Atox1 induction compared to the same subline without irradiation (Fig 4A).

To validate these findings at the transcriptional level, we performed real-time PCR. Irradiation with a dose of 10 Gy resulted in a 4- to 6-fold increase in TP53 gene expression relative to the control. Conversely, Atox1 exhibited weak expression levels. Interestingly, the absence of TP53 led to the activation of Atox1, and radiation further enhanced this effect, particularly in HCT116 cells with TP53$^{-/-}$, where Atox1 expression was approximately 3–4 times higher than in the intact control (Fig 4B). Fluorescent microscopy with the distribution of Atox1 protein after exposure to ionizing radiation at a dose of 10 Gy is shown in the S1 Fig.

The next experiments showed that in the absence of p53 the Atox1 protein can be induced by DNA-damaging agents (doxorubicin and bleomycin) but respond poorly to ROS exposure ($H_2O_2$, ionizing radiation). This effect is observed both at the transcriptional and translational levels. In addition, the Atox1 induction caused by the activation of PKC by the addition of PMA is an important observation. The specific role of Atox1 in response to these stimuli, as well as participation in the regulation of survival and adaptation processes, remains to be established. In our next experiments, we used siRNA transient gene knockdown to identify the effect of inactivation of genes of interest on cell survival and response to genotoxic stress.

## The influence of p53 on Atox1 activity is unidirectional

Cell culture conditions can induce significant changes in cell metabolism and gene expression upon permanent gene inactivation, impacting cell cycle regulation and viability [51]. To avoid these specific changes, we utilized siRNA-mediated knockdown or small molecule inhibitors for transient gene inactivation, allowing us to study the immediate effects of ATOX1, TP53, and their co-inactivation on their reciprocal regulation, as well as changes in cell viability and cell cycle.

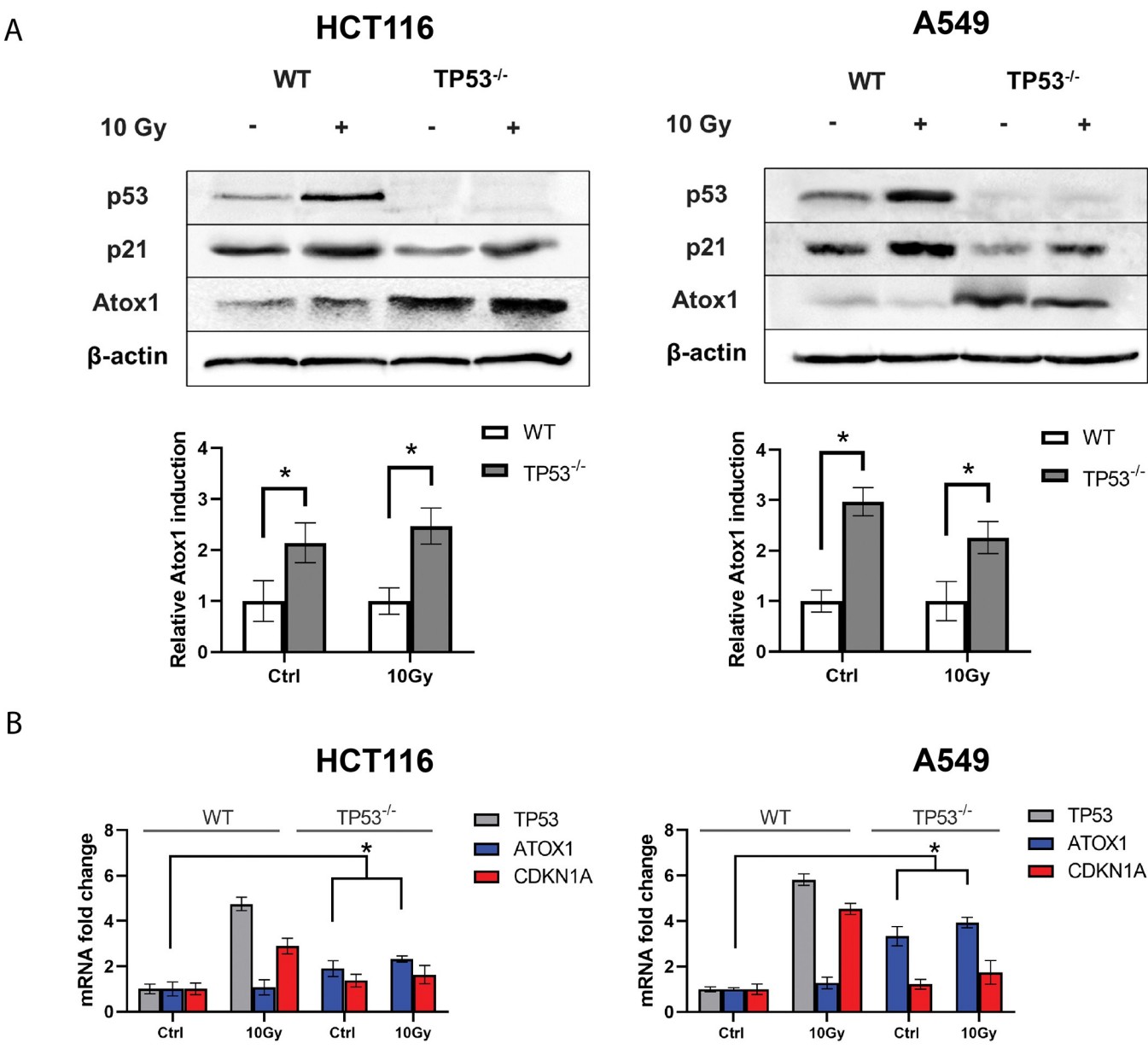

**Fig 4. Influence of ionizing radiation on the activity of Atox1 at different status (WT and KO) of the *TP53* gene in A549 and HCT116 cell lines, 24h after ionizing irradiation (10Gy) exposure. A**—immunoblotting with antibodies to p53, p21, and Atox1; β-actin was used as a normalization. A densitometric analysis of the obtained data is shown below. **B**–RT-qPCR analysis with primers for the *TP53*, *CDKN1A*, and *ATOX1* genes; the *GAPDH* gene was used as a reference. The value of WT 0Gy (control) was taken as 1.0 for all genes and is not shown in the graphs. For all experiments: n = 3, mean +/− SEM, two-way ANOVA, p < 0.05.

To assess knockdown efficiency, we measured *TP53* and *ATOX1* expression levels using RT-qPCR. Our results demonstrated a 10-fold decrease in *TP53* expression and a 100-fold decrease in *ATOX1* expression (Fig 5A). While we previously discussed p53-dependent changes in Atox1 levels, it remained unclear whether Atox1 directly influences p53 activity. To address this, we evaluated the reciprocal regulation of p53 and Atox1 proteins in cells with transient suppression of these genes. Western blot analysis revealed that, similar to HCT116TP53-/- and A549TP53-/- lines, the absence of functional p53 led to increased Atox1

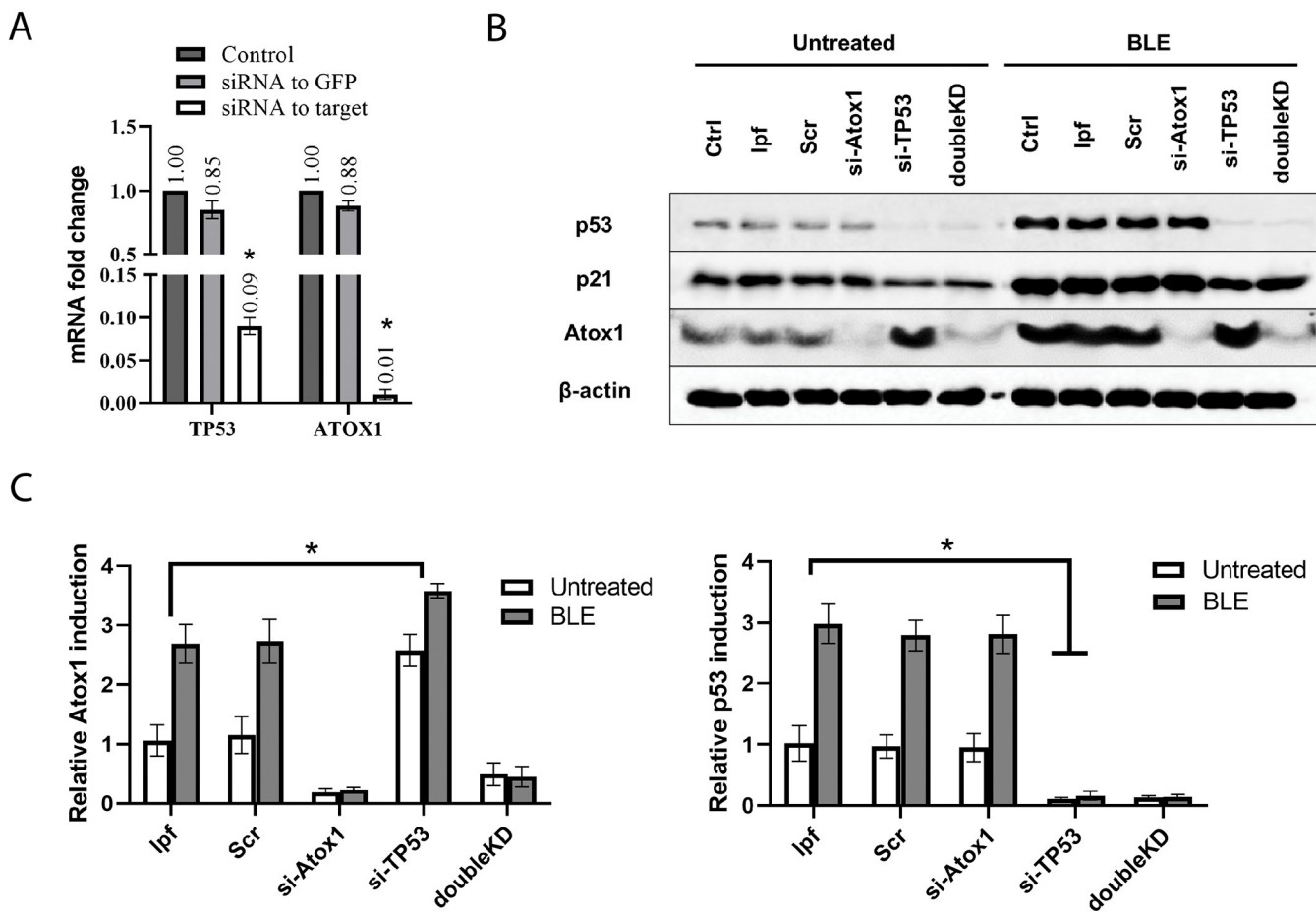

**Fig 5. Analyzing the effect of *ATOX1* and *TP53* gene siRNA-mediated knockdown on mutual expression, A549 cell line.** A—RT-qPCR analysis with primers for *TP53* and *ATOX1* genes after their siRNA knockdown, 24h after inactivation; the *GAPDH* gene was used as a reference. B–immunoblotting with antibodies to p53, p21, and Atox1; β-actin was used as a normalization. *ATOX1* (si-ATOX1), *TP53* (si-TP53), or double *ATOX1/ TP53* (doubleKD) knockdowns were used in the absence (Untreated) and presence (10μM BLE) of bleomycin, 24h after exposure. C—A densitometric analysis of the obtained data is shown below. Controls are taken as 1.0 and not shown on densitometry. For all experiments: n = 3, mean +/− SEM, two-way ANOVA, p < 0.05.

activation (Fig 2). However, Atox1 inactivation did not affect the levels of p53 or p21 (Fig 5B). Simultaneous suppression of both genes (doubleKD) significantly reduced their expression, while some residual Atox1 was observed (~10–20% of control). Addition of bleomycin enhanced protein induction in all samples (Fig 5B). Similar results with the suppression of the *TP53* and *ATOX1* genes were shown by immunoblotting on the HTC116 cell line. The data is shown in the S2 Fig.

## Suppression of *ATOX1* under genotoxic stress increases tumor viability, but simultaneous suppression of *TP53* decreases it

The MTT assay on A549 cells made it possible to assess the viability of cells with active and inactive *ATOX1* or *TP53* when exposed to bleomycin or gamma radiation. The test showed that knockdowns by themselves did not affect cell viability under intact conditions; only double inactivation of the *ATOX1* and *TP53* genes (doubleKD) led to a ~10–12% decrease in survival. On the first day (24 hours) of genotoxic effects, there are also no pronounced changes in cell survival. The addition of the genotoxic drug 10μM BLE or exposure to 10Gy of gamma radiation increased cell death after 72 hours: in the case of ionizing radiation, the survival rate

decreased by 35% compared to the control, and in the case of exposure to bleomycin by 40%. The same is true for samples with lipofectamine (lpf) and scrambled siRNA to *GFP* (Scr). Further, it was found that the frequency of cell death with knockdown of the *ATOX1* gene was reduced on the 3rd day after the respective treatments. Thus, the percentage of surviving Atox1-negative cells after 72 hours after they were exposed to gamma radiation and treatment with bleomycin was 86% and 84.5%, respectively. The control values for wild-type cells after the respective treatments were 72.4% and 67.2%, respectively. At the same time, knockdown of *TP53* reduced cell viability compared to the control: when exposed to radiation and bleomycin, the survival of cells with inactivated *TP53* after 72 hours was 60.9% and 49.2%, respectively. Finally, double gene knockdown resulted in marked cell death: 37.6% and 31.9% on radiation and bleomycin exposure, respectively ([Fig 6A]). Thus, *ATOX1* inactivation serves as a kind of "protector" of cells from death, but with simultaneous inactivation of *TP53*, this property is also removed, and the opposite effect is observed: inhibition of cell survival.

To elucidate the reasons for the observed effects of death avoidance upon inactivation of *ATOX1*, we examined the distribution of cell cycle phases using A549 cell line (as the line that most effectively responds to stressful conditions at the transcriptional level) under the same conditions (*TP53*, *ATOX1*, or knockdown of both genes, with or without bleomycin). The addition of siRNA to *TP53* and *ATOX1* in the case of untreated cells (addition of 250 nM lepofectamine 2000) practically did not change the distribution of $G_1$, S, and $G_2$/M phases after 24–72 hours (sub$G_1 < 10\%$). A different situation was observed with simultaneous knockdown of *TP53* and *ATOX1* (doubleKD): while in the control group the sub$G_1$ phase was 2–5%, the absence of both genes led to an increase in this phase to 10–13%, while there were no noticeable changes in other phases, $G_1$ and $G_2$/M.

In the bleomycin-supplemented group (250 nM lpf 2000, 10 μM BLE), the differences were more pronounced ([Fig 6B]). For example, 24 hours after the addition of bleomycin, about 20% of the cells are in the Sub$G_1$ phase, while after 72 hours the relative number of events in this phase rises to 29%. The addition of *GFP* siRNA (Scr) did not change the ratio between cell cycle phases relative to the controls described above. Suppression of TP53 and, accordingly, its reparative functions and control of cell cycle arrest, did not lead to an obvious increase in sub$G_1$: 18.2% at 24 hours after the addition of bleomycin and 25.5% at 72 hours, respectively. However, at this point, a time-dependent increase in the $G_2$/M phase was observed (23.5% and 40.5%, respectively). Unexpected, but consistent with the logic of the MTT test, changes in the cell cycle were observed when *ATOX1* was suppressed: the transition of cells to the sub$G_1$ fraction slowed down. Thus, 24 hours after the addition of bleomycin, sub$G_1$ (in the group with si-Atox1) was equal to 22% while the values in the control group (lpf) were 17%. However, after 72 hours, sub$G_1$ (si-Atox1) was 16%, with values in the control group (lpf) of 28%. Finally, in the BLE group with inactivation of both genes (doubleKD), increased tumor cell death is observed with almost complete escape of cells from the $G_2$/M phase. For example, 24 hours after the addition of bleomycin, the sub$G_1$ and $G_2$/M phases were 31.3% and 18.8%, respectively. After 72 hours, these figures were 42% and 12.5%, respectively. According to our results, active Atox1 in cells with DNA damage can induce apoptosis, but the absence of its functioning form creates a block at the $G_1$/S checkpoint and limits the ability of cells to go into apoptosis (sub$G_1$). Disabling the second gene, *TP53*, allows cells to bypass this effect and successfully redistribute into the sub$G_1$ phase.

## Discussion

In this study, we established that the expression of the transcription factor and antioxidant protein Atox1 is more pronounced in cell lines with inactivated *TP53*. Specifically, we observed

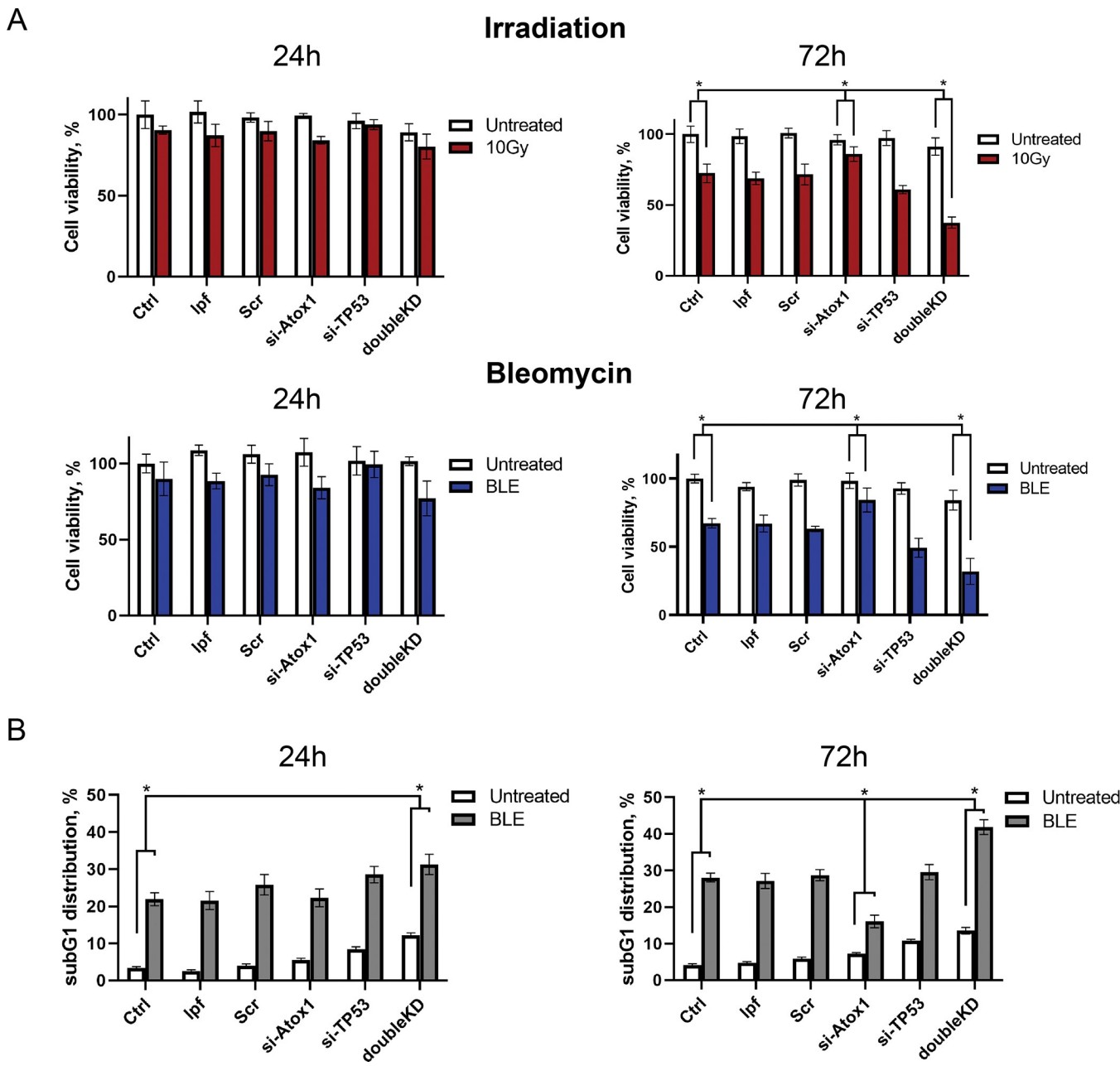

**Fig 6. Influence of *ATOX1*, *TP53*, and double knockdown on cell viability in the absence and presence of genotoxic stress in A549 cell line.** A–MTT-assay after *ATOX1* (siATOX1), *TP53* (siTP53) or double *ATOX1/ TP53* (doubleKD) knockdowns in the absence (Untreated) and presence of ionizing irradiation (10Gy) and 10μM bleomycin (BLE) at the different time points: 24 or 72 hours after exposure. B–subG$_1$ phase accumulation is caused by *TP53* (siTP53), *ATOX1* (siATOX1), or double genes (doubleKD) knockdowns in the absence (Untreated) and presence of 10μM bleomycin (BLE). For all experiments: n = 3, mean +/− SEM, two-way ANOVA, $p < 0.05$.

that cell lines with wild type p53, such as HCT116 and A549, exhibited reduced gene expression and induction of Atox1 protein compared to cells with inactive p53, either through knockout or knockdown techniques. Furthermore, common antitumor drugs such as doxorubicin and bleomycin, as well as exposure to therapeutic doses of ionizing radiation, activate p53 in normal cells, while the concentration of Atox1 protein is significantly increased only in cells with inactive *TP53*. Conversely, the suppression of the *ATOX1* gene using small

interfering RNAs (siRNAs) did not result in any transcriptional or translational changes in the level of *TP53* in these cell lines. These results suggest that the p53 tumor suppressor acts as a negative regulator of Atox1, while no reciprocal feedback mechanism was identified. Fluorescent microscopy with antibodies to Atox1 and subcellular fractionation analysis revealed that the presence of genotoxic stimuli caused only minor translocation of Atox1 in the nucleus of *TP53*$^{-/-}$ cells, while in WT cells level of intranuclear presence of Atox1 increases upon genotoxic stimuli. Differences in microscopy data and western blot might suggest that intranuclear localization occurs with simultaneous increase of expression level (as was shown by qPCR and western blot data) hence differences between cytoplasmic and nuclear level of Atox1 is less pronounce. Moreover, translocation in WT and *TP53*$^{-/-}$ cells could indicate that p53 is necessary for Atox1 nuclear translocation, which should be investigated in further research.

Moreover, our study examined the impact of siRNA-mediated knockdowns of the *TP53* and *ATOX1* genes on cell survival and cell cycle distribution in the absence of cytotoxic drugs. It was found that knockdown of these genes did not alter these parameters. When treated with bleomycin, both cell lines exhibited an increased accumulation of cells in the subG$_1$ phase (an indicator of cell death), which was further enhanced with *TP53* inactivation through siRNA knockdown. Surprisingly, we observed a decrease in the subG$_1$ phase accumulation when bleomycin was added to cells with inactivated *ATOX1*. These findings were corroborated by the MTT assay. Lastly, we investigated the simultaneous knockdown of both *ATOX1* and *TP53* genes, which resulted in an increased apoptosis rate compared to cells with inactive *TP53* alone. This effect was approximately two-fold higher 72 hours after drug exposure, while the number of G$_2$/M-arrested cells decreased. This intriguing observation presents a paradoxical scenario, whereby inactivation of *ATOX1* protects cells from death, but additional suppression of *TP53* enhances the apoptotic effect by abolishing the G$_2$/M cell cycle block and promoting cell death in the subG$_1$ phase.

Collectively, these findings suggest the existence of a potential mechanism by which *ATOX1* is inversely associated with p53 levels and facilitates cell death rather than, as previously proposed, cell survival by eliminating ROS [32, 33, 36, 37]. Moreover, in our experiments, we did not observe a significant increase of Atox1 level in response to H$_2$O$_2$ treatment, suggesting the primal role of Atox1 as a transcription factor; however, the underlying cause of this phenomenon warrants investigation. While the role of p53 in governing the balance between repair and apoptosis in tumor cells has been extensively studied over the course of more than four decades [52–54], the precise involvement of Atox1 in this context remains enigmatic. Several evidences indicate that Atox1 can positively regulate the expression of cyclin D1, a key factor in cell cycle progression, and the transition from the G$_1$ to S phase [33, 37, 55]. It is plausible to speculate that inhibiting Atox1 upon exposure to DNA-damaging agents, such as chemotherapy or radiotherapy, prevents cells from bypassing the G$_1$/S checkpoint with genomic damage (Fig 7A). In the absence of Atox1-mediated CCND1 expression, cells fail to accumulate critical damage during DNA replication and do not undergo mitotic catastrophe [56]. This leads to a reduction in the subG$_1$ and G$_2$/M phases in ATOX1 knockdown experiments (Fig 6). Moreover, experimental data also elucidate the role of p53 as a negative regulator of Atox1: in TP53-inactivated cells, Atox1 expression is elevated, resulting in tumor cell death due to bypassing the G$_1$/S and G$_2$/M checkpoints without proper DNA repair (Fig 6B).

Nevertheless, this theory does not fully explain the synergistic impact of *ATOX1/TP53* double knockdown. For instance, if the translocation of Atox1 into the nucleus was not conclusively observed during the experiment, how does it regulate cyclin D1? Is Atox1 involved in proapoptotic signaling, the inhibition of which protects cells from death? Could this be linked to the purported ability of Atox1 to function as a non-canonical modulator for the MAPK cascade, specifically mediating the phosphorylation of the transcription factor Erk (Ras-ERK signaling

## Under genotoxic stress

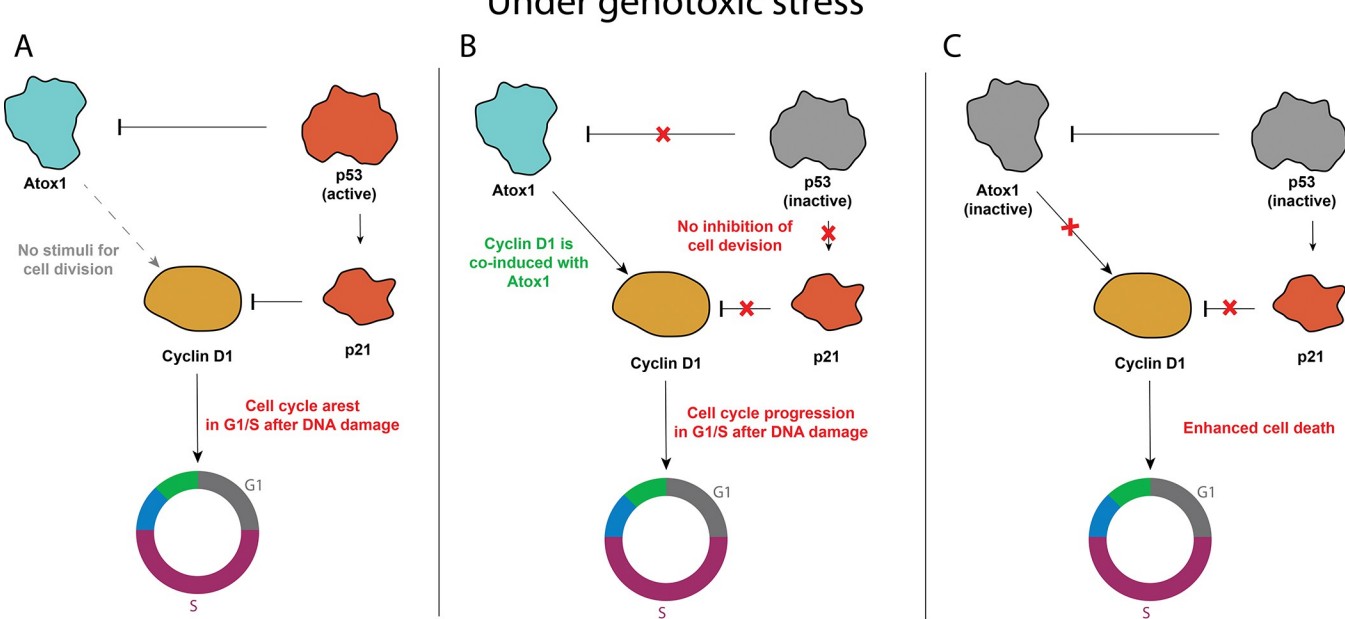

**Fig 7. Hypothetical concept of Atox1-mediated cell cycle regulation via cyclin D1 and p53 influence on the process under genotoxic stress.** A–when p53 is fully functional, Atox1 is suppressed as well as cyclin D1 (via p21 activation), B–when p53 is inactivated, Atox1 is released from suppression and promotes cell cycle progression due to the induction of cyclin D1, C–In case of double knockdown, when both Atox1 and p53 are suppressed, cells cannot properly regulate the transition from one phase of the cell cycle to another, resulting in increased sensitivity to DNA damaging agents.

pathway, MAPK/ERK) [27, 57, 58]? In this scenario, the p53-controlled Atox1-mediated regulatory network involving CCND1 may be even more intricate and context-dependent. Recent work about cuproptosis describes p53 participation in the regulation of this process [59] which may indicate that copper-mediated cell death is possibly realized via Atox1. This may indicate broader functions of Atox1 in cellular signaling related to cell survival. In any case, further investigations are warranted to unravel the role of Atox1-p53 in the regulation of Cyclin D1.

## Conclusion

In conclusion, the present research lays the foundation for establishing a comprehensive and coherent framework for understanding the relationship between copper metabolism proteins and p53 activity in cell malignancy. This research opens the door for future studies to explore the role of Atox1 and p53 interactions in tumor progression and potential approaches to cancer therapy by targeting these proteins. Future investigations should focus on elucidating the underlying mechanisms of this interaction, including the involvement of Cyclin D1, p63 and p73, Ras-ERK, and other proteins that regulate the $G_1$/S and $G_2$/M transitions. The identification of these factors and their association with p53 holds great promise for both diagnostic and therapeutic purposes. A deep and detailed study of these interactions in tumors of different localizations under the influence of antitumor drugs and ionizing radiation agents will allow the development of optimal combined schemes for the treatment of tumors.

## Materials and methods

### Cell lines and reagents

Transformed human cell lines were used: HCT116 (colon adenocarcinoma) with intact p53; HCT116p53$^{-/-}$ with a deletion of both alleles of the *TP53* genes, as well as the A549 line with

wild (A549) and knockout p53 (A549p53$^{-/-}$) by the CRISPR-Cas9 method, acquired at ATCC. The cells were cultured in Dulbecco's modified Eagle's medium (DMEM, Biolot, Russia) supplemented with 2 mM L-glutamine, 10% fetal bovine serum (PAA, USA), and 100 U/mL gentamicin (Biolot). Only cells in the logarithmic growth phase, with no more than 15 passages, were used in the experiments. All other reagents used in this study were obtained from Sigma, USA, unless otherwise specified.

## Compounds and ionizing radiation

Antitumor and cytotoxic compounds for DNA damage induction—doxorubicin, bleomycin, phorbol 12-myristate 13-acetate (PMA)—were used at concentrations corresponding to the IC50 for specific lines.

For irradiation of tumor cells with gamma photons, a radiotherapy unit named RUM-17 was used, provided for work by the Military Medical Academy named after S.M. Kirov. A preselected therapeutic dose of 10 Gray (Gy) was used in the experiments. The irradiation parameters included a voltage across the tube of 180 kV, a current of 10 mA, a focal length of 50 cm, a 1-mm Al filter, a 0.5-mm Cu filter, and a dose rate of 0.32 Gy/min.

## Cell viability analysis

To study the effect of bleomycin and 10Gy ionizing radiation on cellular metabolic activity in the condition of *TP53*, *ATOX1*, and both genes inactivation, the MTT assay was used [60]. The number of surviving cells was determined by the optical density of a solution of reduced MTT (3-(4,5-dimethyl-2-thiazolyl)-2,5-diphenyl-2H-tetrazolium bromide) dye with NADP-H-dependent oxidoreductases at a wavelength of 570 nm.

## Cell cycle assay

The distribution of cell cycle (according to DNA ploidy) was analyzed on a CytoFlex B2-R2-V0 flow cytometer (USA) in PE or Rhodamine channels. A 2D PE-W versus PE-A plot was used to exclude cell conglomerates. 20,000 events were accumulated for each sample. The indicators were analyzed in the areas SubG1, G1, and G2/M.

## Reverse transcription

Isolation of total RNA from cells was performed using the Total RNA isolation protocol with ExtractRNA buffer (Evrogen, Russia) according to the manufacturer's protocol. cDNA was generated from total RNA (2 μg) by using MMLV reverse transcriptase (Evrogen, Russia). Reverse transcription PCR reaction conditions were as follows: 25˚C-10 min, 42˚C-50 min, 70˚C-10 min, 10˚C-10 sec.

## Real-time PCR analysis

For real-time PCR, a mixture for PCR was prepared, which included: a mixture of 5x qPCR SYBR Green I (Evrogen); forward and reverse primers, 10 μM each; nuclease-free H2O. A negative control was prepared: a sample without the addition of the corresponding cDNA. Amplification conditions:

- Stage 1 (1 cycle): 94˚C-3 min, 60˚C-40 sec, 72˚C-40 sec

- Stage 2 (28–30 cycles): 94˚C-10s, 60˚C-10s, 72˚C-20s

- Stage 3 (1 cycle): 72˚C-3 min

**Table 1. List of PCR primers.**

| Gene | Primer | Sequence |
|---|---|---|
| TP53 | Forward | 5'-GAGCTGAATGAGGCCTTGGA-3' |
| | Reverse | 5'-CTGAGTCAGGCCCTTCTGTCTT-3' |
| ATOX1 | Forward | 5'-TCTGAGCACAGCATGGACACTC-3' |
| | Reverse | 5'-TCTGGAAGCCAGCGGGAGGAT-3' |
| CDKN1A | Forward | 5'-AGTCAGTTCCTTGTGGAGCC-3' |
| | Reverse | 5'-CATTAGCGCATCACAGTCGC-3' |
| GAPDH | Forward | 5'-CAGTCAGCCGCATCTTCTTTTGCGTCG-3' |
| | Reverse | 5'-CAGAGTTAAAAGCAGCCCTGGTGACCAGG-3' |
| HPRT1 | Forward | 5'-TATATCCAACACTTCGTGGGGTC-3' |
| | Reverse | 5'-ACAGGACTGAACGTCTTGCTC-3' |

- Stage 4 (storage): 4°C

After the completion of the reactions, the expression of the products was determined by the ΔCt method, where Ct (threshold cycle) is the cycle at which the fluorescence level reaches a certain value (preselected threshold), and Δ is the change in the expression of the gene of interest relative to the reference gene, which is selected as normalization. In the experiment, transcripts of the *GAPDH* and *HPRT* genes were used for normalization. In all groups, differences from the control were significant at $p \leq 0.05$ (one-way ANOVA test).

The primers used are listed in Table 1.

## siRNA transfection

Lipofectamine 2000 (Invitrogen) was used to transfect siRNAs according to the manufacturer's instructions in OptiMEM media. Transfection of siRNA was carried out 24 hours before treatment with DNA damage drugs or ionizing radiation using 250 pmol of siRNA. *GFP* sequences were used as scrambled RNA.

The siRNAs used are listed in Table 2.

## Western blotting

Protein electrophoresis was conducted using a polyacrylamide (PAGE) gel containing 10% SDS. A total of 35 μg of total protein was added to the gel lanes. Following electrophoresis, the proteins were transferred to a nitrocellulose membrane (Amersham, USA) using Tris-Glycine buffer. The membranes were then incubated overnight at 4°C with primary antibodies targeting p53, p21, and Atox1 proteins (Abcam, diluted 1:500–1:2000 in TBST). Anti-β-actin antibodies, diluted 1:1000, were used as an internal control. Protein visualization was achieved through chemiluminescence using secondary antibodies specific to mouse or rabbit IgG

**Table 2. List of siRNAs.**

| Gene | siRNA | Sequence |
|---|---|---|
| TP53 | Sense | GGAAGACUCCAGUGGUAAUCUdTdT |
| | Antisense | AGAUUACCACUGGAGUCUUCCdTdT |
| ATOX1 | Sense | GAAGGUCUGCAUUGAAUCUGAdTdT |
| | Antisense | UCAGAUUCAAUGCAGACCUUCdTdT |
| GFP | Sense | GCAAGCUGACCCUGAAGUUdTdT |
| | Antisense | AACUUCAGGGUCAGCUUGCdTdT |

(Amersham, USA) conjugated with horseradish peroxidase. Secondary antibody dilutions ranged from 1:2000 to 1:5000. Detection was performed utilizing the ChemiDoc Touch gel-documentation system (BioRad). Densitometry analysis to evaluate the relative protein content was conducted using the Grey Mean Value Calculation tool in the ImageJ program.

### Immunofluorescence staining

Immunofluorescence staining was performed by fixing cells with 4% paraformaldehyde (PFA), permeabilizing them with 0.2% Triton X, and blocking with 1% bovine serum albumin. Cells were then incubated overnight at 4˚C with primary antibodies targeting Atox1 (Abcam, diluted 1:300). Subsequently, cells were incubated with Alexa Fluor-conjugated secondary antibodies (Thermo Fisher Scientific) for 1 hour at room temperature. Nuclear staining was achieved using DAPI. Images were captured using the fluorescence microscope Leica DMi8.

### Subcellular fractionation

To determine the intracellular distribution of Atox1 and p53 proteins under the treatment of cytostatic drugs was used the subcellular fractionation method proposed by Yu Z. [61], based on centrifugation of the cell fraction in a solution of 250 mM Sucrose and 20 mM HEPES, which allows to separate cell lysates into 2 fractions: cytoplasmic and nuclear.

### Statistical methods

Prism 8 (GraphPad) was used for statistical analysis. For the results of cell culture and immunostaining experiments, Student's t-test was used to calculate P values. Mean ± standard error of the means (SEMs) is shown in the figures. Differences were considered significant if $p < 0.05$.

### Supporting information

**S1 Fig. Influence of ionizing radiation on the activity of Atox1 at different status (WT and KO) of the TP53 gene in A549 and HCT116 cell lines, 24h after ionizing irradiation (10Gy) exposure.** Immunofluorescence staining with primary antibodies to Atox1 and secondary antibodies with AlexaFluor488. DAPI was used for nuclei staining. TP53$^{-/-}$–cells without TP53. For all experiments: n = 3, mean +/− SEM, two-way ANOVA, p < 0.05. (TIFF)

**S2 Fig. Analyzing the effect of ATOX1 and TP53 gene siRNA-mediated knockdown on mutual expression, HCT116 cell line.** Immunoblotting with antibodies to p53, p21, and Atox1; beta-actin was used as a normalization. ATOX1 (si-ATOX1), TP53 (si-TP53) or double ATOX1/ TP53 (doubleKD) knockdowns were used in the absence (Untreated) and presence (10μM BLE) of bleomycin, 24h after exposure. (TIFF)

**S1 Raw images.** (PDF)

### Acknowledgments

The authors express their deep gratitude to the scientific consultant, MD Alexander Shtil, for mentoring and advice on setting up experiments and interpreting the results.

## Author Contributions

**Conceptualization:** Sergey Tsymbal, Oleg Kuchur.

**Data curation:** Sergey Tsymbal.

**Formal analysis:** Oleg Kuchur.

**Funding acquisition:** Oleg Kuchur.

**Investigation:** Sergey Tsymbal, Aleksandr Refeld, Viktor Zatsepin.

**Methodology:** Aleksandr Refeld, Viktor Zatsepin.

**Project administration:** Oleg Kuchur.

**Supervision:** Oleg Kuchur.

**Validation:** Sergey Tsymbal, Aleksandr Refeld.

**Visualization:** Sergey Tsymbal, Aleksandr Refeld.

**Writing – original draft:** Sergey Tsymbal, Aleksandr Refeld, Oleg Kuchur.

**Writing – review & editing:** Viktor Zatsepin, Oleg Kuchur.

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
