## [Decision Letter · Decision Letter 0]

13 Nov 2023

PONE-D-23-26090The p53 Protein is a Suppressor of Atox1 Copper Chaperon in Tumor Cells Under Genotoxic EffectsPLOS ONE

Dear Dr. Tsymbal,

Thank you for submitting your manuscript to PLOS ONE. After careful consideration, we feel that it has merit but does not fully meet PLOS ONE’s publication criteria as it currently stands. Therefore, we invite you to submit a revised version of the manuscript that addresses the points raised during the review process.

Overall, your manuscript will be acceptable for publication if you address Reviewer #2's Major Point 1, and their minor points 1-3.  You would have a stronger case if you conduct the suggested work to support your immunofluorescence data of nuclear activity, and in the absence of this may wish to provide any further support of the immunofluorescence technique from the literature. With regard to the R3 minor point 3, your chief area to address about colloquial language is the second paragraph in the section "Atox1 is induced in a p53-dependent manner during genotoxic stress."We look forward to receiving your revised manuscript.

We look forward to receiving your revised manuscript.

Kind regards,

Roy P. Planalp, Ph.D.

Academic Editor

PLOS ONE

“YES

This research was funded by Russian Science Foundation, grant number 22-24-0058.”

Reviewers' comments:

Reviewer's Responses to Questions

**Comments to the Author**

1. Is the manuscript technically sound, and do the data support the conclusions?

Reviewer #1: Yes

Reviewer #2: Yes

2. Has the statistical analysis been performed appropriately and rigorously? 

Reviewer #1: Yes

Reviewer #2: Yes

3. Have the authors made all data underlying the findings in their manuscript fully available?

Reviewer #1: Yes

Reviewer #2: Yes

4. Is the manuscript presented in an intelligible fashion and written in standard English?

Reviewer #1: Yes

Reviewer #2: Yes

5. Review Comments to the Author

Reviewer #1: The connection between copper metabolism and progress of cancer is still not known. Although several studies pointed out on this link, there are a lot of missing information regarding the mechanism. In this manuscript, the authors explored the connection between p53 and ATOX1 expression using cell experiments and knockdown of the two proteins. Moreover, the effect of cytotoxic drugs and irradiation on the expression level was also explored. To my best of knowledge, this study was well conducted, and the findings are very interesting, I therefore recommend accepting the manuscript in its current version.

Reviewer #2: In this manuscript, Tsymbal et al. examine the relationship between the tumor suppressor p53 and the copper chaperone Atox1 in two different cancer cell lines (A549 and HCT116) exposed to various types of genotoxic stress. The authors establish that an inverse correlation exists between p53 and Atox1 at both the transcriptional and translational levels. Moreover, they demonstrate that suppression of both genes increases apoptosis, but reduced expression of Atox1 alone decreases apoptosis when cells are treated with chemotherapeutic agents. Overall, the data are convincing, the experiments are well designed and controlled, and for the most part, the conclusions are supported by the data. Below are some comments/suggestions that should be addressed in a revised version of the manuscript.

Major Points:

1) While a MTT assay is valid for determining cell viability, it is not an appropriate method to assess apoptosis per se. The experiments in Figure 5A should be repeated using a validated assay to specifically measure apoptosis (Annexin V/PI staining, cleaved caspase-3 or cleaved PARP Western blotting, etc.)

2) In Figures 1C and 2C, the lack of nuclear translocation of Atox1 as demonstrated by immunofluorescence microscopy is striking. However, these results would be strengthened by Western blotting of nuclear vs. cytosolic fractions in a subcellular fractionation assay to confirm.

Minor Points:

1) In Figure 3B, statistical significance is not indicated. This should be included.

2) PMA activates protein kinase C (PKC) and, in turn, many downstream signaling pathways. The focus on the NF-kappaB pathway is not well justified.

3) The manuscript is written in what seems to be a colloquial or conversational manner. While there are no major grammatical errors, it should be edited so that it conforms to standard scientific writing.

6. PLOS authors have the option to publish the peer review history of their article (what does this mean?). If published, this will include your full peer review and any attached files.

Reviewer #1: No

Reviewer #2: No

---

## [Author Response · Author response to Decision Letter 0]

29 Nov 2023

The authors express their gratitude for the comprehensive review conducted on our manuscript titled: "The p53 Protein Functions as a Suppressor of Atox1 Copper Chaperone in Tumor Cells Under Genotoxic Effects." We have carefully revised the manuscript and are pleased to present the updated version of our work.

In addition to addressing grammatical and stylistic errors, we have also made revisions to ensure that the language used throughout the manuscript adheres to academic standards. The English language in the second paragraph of the section "Atox1 is induced in a p53-dependent manner during genotoxic stress" has been corrected in accordance with academic conventions. Furthermore, the manuscript has been edited to comply with the formatting requirements of PLOS ONE, including the proper naming of paragraphs and chapters, formatting of the list of references, and labeling of files. Additionally, all gene names mentioned in the text have been italicized, and the same style has been applied to all figures.

Second, an explanation to the source of funding is added: “The funders had no role in study design, data collection and analysis, decision to publish, or preparation of the manuscript”.

Third, statistical significance is added to Figures 3B and 4C. Also a PDF file combining all the original Western blot images was uploaded as required by the journal.

We would like also respond to the comments of the reviewers:

Reviewer 1. The connection between copper metabolism and progress of cancer is still not known. Although several studies pointed out on this link, there are a lot of missing information regarding the mechanism. In this manuscript, the authors explored the connection between p53 and ATOX1 expression using cell experiments and knockdown of the two proteins. Moreover, the effect of cytotoxic drugs and irradiation on the expression level was also explored. To my best of knowledge, this study was well conducted, and the findings are very interesting, I therefore recommend accepting the manuscript in its current version.

Response. Authors are thankful for the kind reply of the reviewer. We glad that our work was appreciated to such great extent.

Reviewer 2. In this manuscript, Tsymbal et al. examine the relationship between the tumor suppressor p53 and the copper chaperone Atox1 in two different cancer cell lines (A549 and HCT116) exposed to various types of genotoxic stress. The authors establish that an inverse correlation exists between p53 and Atox1 at both the transcriptional and translational levels. Moreover, they demonstrate that suppression of both genes increases apoptosis, but reduced expression of Atox1 alone decreases apoptosis when cells are treated with chemotherapeutic agents. Overall, the data are convincing, the experiments are well designed and controlled, and for the most part, the conclusions are supported by the data. Below are some comments/suggestions that should be addressed in a revised version of the manuscript.

Major Points: 

1) While a MTT assay is valid for determining cell viability, it is not an appropriate method to assess apoptosis per se. The experiments in Figure 5A should be repeated using a validated assay to specifically measure apoptosis (Annexin V/PI staining, cleaved caspase-3 or cleaved PARP Western blotting, etc.).

Response. Authors are grateful for reviewer’s detailed analysis of present work and valuable comments. Indeed, MTT assay could not be used to determine the mechanism of cell death, however in this experiment we were interested in determining rather quantitative than qualitative characteristics of cell death process. The main idea was to characterize how differences in the expression levels of the ATOX1 and TP53 genes can affect overall cell viability and survival when genotoxic drugs are added. In addition, we performed cell cycle analysis to further understand the changes underlying the cell death process. Studying the mechanism of cell death is also an important and relevant point for research that is planned to be carried out in the next work.

2) In Figures 1C and 2C, the lack of nuclear translocation of Atox1 as demonstrated by immunofluorescence microscopy is striking. However, these results would be strengthened by Western blotting of nuclear vs. cytosolic fractions in a subcellular fractionation assay to confirm. 

The authors performed additional experiments suggested by the reviewer. The results of subcellular fractionation (WB for the nuclear and cytoplasmic fractions of cells) have been added; these data are combined with fluorescence microscopy data for Atox1 staining in a separate Figure 3. Corresponding changes have been made to the chapter “Materials and Methods”. New text has been added describing the results (lines 189-199), as well as a discussion (lines 345-353).

Minor Points: 

1) In Figure 3B, statistical significance is not indicated. This should be included. 

Statistical significance is added to the Figures 4B and 5C which correspond to the Figures 3B and 4C in the old version of the manuscript.

2) PMA activates protein kinase C (PKC) and, in turn, many downstream signaling pathways. The focus on the NF-kappaB pathway is not well justified. 

Fragments of text about the role of the PMA compound as an activator of the NF-kB pathway have been rewritten (lines 137-148). Now the focus is on a more relevant cascade: PKC-MAPK1-Atox1.

3) The manuscript is written in what seems to be a colloquial or conversational manner. While there are no major grammatical errors, it should be edited so that it conforms to standard scientific writing.

The authors thank the reviewer for his valuable feedback. We have rewritten the text and corrected major grammatical errors to meet academic standards.

---

## [Editor Report · Decision Letter 1]

4 Dec 2023

The p53 Protein is a Suppressor of Atox1 Copper Chaperon in Tumor Cells Under Genotoxic Effects

PONE-D-23-26090R1

Dear Dr. Tsymbal,

We’re pleased to inform you that your manuscript has been judged scientifically suitable for publication and will be formally accepted for publication once it meets all outstanding technical requirements.

Kind regards,

Roy P. Planalp, Ph.D.

Academic Editor

PLOS ONE

Additional Editor Comments (optional):

It appears that all reviewer concerns have been adequately addressed.  Studies conducted are sufficient, and I commend your work. 
---

## [Editor Report · Acceptance letter]

7 Dec 2023

PONE-D-23-26090R1 

The p53 protein is a suppressor of Atox1 copper chaperon in tumor cells under genotoxic effects 

Dear Dr. Tsymbal:

I'm pleased to inform you that your manuscript has been deemed suitable for publication in PLOS ONE. Congratulations! Your manuscript is now with our production department. 

Kind regards, 

on behalf of

Dr. Roy P. Planalp 

Academic Editor

PLOS ONE